# Biologic Agents in Idiopathic Hypereosinophilic Syndrome [note 1]

**DOI:** 10.3390/ph18040543

**Published:** 2025-04-08

**Authors:** Ourania Papaioannou, Fotios Sampsonas, Panagiota Tsiri, Vasilina Sotiropoulou, Ioannis Christopoulos, Dimitrios Komninos, Argyrios Tzouvelekis

**Affiliations:** Department of Respiratory Medicine, University Hospital of Patras, 26504 Patras, Greece; ouraniapapaioannou@outlook.com (O.P.); fsampsonas@gmail.com (F.S.); tsiripanayiota@gmail.com (P.T.); v.swtiropoulou@gmail.com (V.S.); christopoulosi94@gmail.com (I.C.); komninos312@gmail.com (D.K.)

**Keywords:** hypereosinophilic syndrome, biologic agents, eosinophils

## Abstract

**Background:** Hypereosinophilic syndrome (HES) is a heterogeneous group of rare disorders defined by the presence of marked eosinophilia resulting in end organ damage. The diagnostic approach is multidisciplinary and treatment goals include reductions in flares and eosinophils with minimal drug-related side effects. **Results:** Eleven patients (*n* = 11) with a diagnosis of idiopathic HES were included in the study [M/F: 6/5, median age: 54 (95% CI: 38.2 to 68.5), smokers/never smokers: 5/6]. Asthma was present in the majority of them (*n* = 8, 72.7%); four patients (*n* = 4, 36.4%) presented with eosinophilic pleural effusions, two patients (*n* = 2, 18.2%) with cardiac arrhythmias, and one with bilateral eyelid angioedema. Eight patients (72.7%) were treated with mepolizumab (300 mg/month) and three (27.3%) with benralizumab (30 mg/4 weeks). The median values of eosinophils at baseline and 12 months after initiation of biologic agent were 3000 (95% CI: 2172 to 11,365) K/μL and 50 (95% CI: 3 to 190) K/μL, respectively, *p* = 0.0002. All patients with concomitant asthma (*n* = 8) experienced elimination of asthma flares, asthma control (ACQ < 0.75), functional improvement (mean ΔFEV1: 857 ± 594 mL), and an 82% reduction in oral corticosteroids, *p* = 0.0001. **Materials and Methods:** Patients with highly characterized idiopathic HES treated with anti-eosinophilic agents between 1 October 2019 and 1 October 2023 were retrospectively included in the study. The aim of this study was to present clinical, laboratory, and functional features and outcomes in patients with thoroughly investigated idiopathic HES treated with biologic agents targeting eosinophils. **Conclusions:** Biologic agents in patients with idiopathic HES—following thorough diagnostic investigation—are both safe and effective, sparing the toxicity of immunosuppressive agents. Real-life data from larger registries are greatly anticipated.

## 1. Introduction

Hypereosinophilic syndrome (HES) is a heterogeneous group of rare disorders characterized by the presence of remarkable eosinophilia, resulting in end organ damage if treatment is not timely and appropriate [1,2]. In order to diagnose idiopathic HES, we need to exclude all primary and secondary causes of hypereosinophilia [3]. The current definition of HES requires persistent eosinophilia with a corresponding absolute eosinophil count of >1.5 × 10^9^/L for a shorter period of 2 to 4 weeks (compared to the historical criterion of more than 6 months outlined by Chusid and colleagues in 1975), and concomitant tissue damage [4,5]. More specifically, the diagnostic criteria for idiopathic HES include sustained hypereosinophilia (eosinophil count ≥ 1.5 × 10^9^/L) recorded more than once in a minimum time interval of two weeks, ≥10% eosinophilia, organ failure attributable to the eosinophilia or tissue infiltration by eosinophils, and exclusion of reactive eosinophilia, lymphocyte-variant HES, chronic eosinophilic leukemia, and other myeloid or lymphoid neoplasm [4]. According to the World Health Organization classification for eosinophilic disorders, the diagnosis of idiopathic HES may be in the context of working diagnosis until the final cause of eosinophilia is determined. It is important to mention that in the era of multiple molecular markers of neoplasia, the percentage of patients diagnosed with idiopathic HES decreased [6]. Despite the fact that the epidemiology of HES is not well characterized, the estimated prevalence of HES is up to 6.3 per 100,000 person-years globally [7].

Clinical manifestations are highly variable depending on specific organ involvement. The most common symptomatology involved in HES includes cutaneous (e.g., rash, urticaria, angioedema), pulmonary (e.g., shortness of breath, cough, pulmonary infiltrates, pleural effusion), gastrointestinal (e.g., abdominal pain, nausea, vomiting), constitutional (e.g., fever, fatigue), cardiovascular (e.g., myocardial ischemia, arrhythmias), and neurological (e.g., numbness, weakness, muscle atrophy) manifestations. Among the signs and symptoms recorded, the most common were weakness, fatigue, and cough in 25% of patients, followed by dyspnea, myalgias or angioedema, rash or fever, and rhinitis [8]. Persistent eosinophilia might potentially involve every organ. Skin manifestations were the most common, reported in 69% of patients, followed by respiratory (44%) and gastrointestinal (38%) manifestations. Cardiac disease related to underlying eosinophilic disease was eventually identified in 20% of patients. It is notable that endocardial damage can lead to mural thrombi, increased embolic risk, and restrictive cardiomyopathy [9].

The main pillars of diagnostic approach are the exclusion of secondary reactive causes of eosinophilia, assessing end organ damage based on signs and symptoms, and evaluation for primary clonal eosinophilia [3]. With regard to the secondary causes of eosinophilia, a multidisciplinary discussion with subspecialty experts seems crucial in order to exclude infections, such as parasitic (strongyloides, toxocara, schistosoma, echinococcus, entamoeba, cystoisospora, ascaris, hookworm, trichinella, paragonimus, clonorchis, filariasis), viral (human immunodeficiency virus), fungal (coccidiodes, histoplasma, cryptococcus, pneumocystis), mycobacterial (tuberculosis), and bacterial infections; allergies, such as asthma, allergic rhinitis, atopic dermatitis, and allergic bronchopulmonary aspergillosis; autoimmune causes, such as inflammatory bowel disease, celiac disease, eosinophilic granulomatosis with polyangiitis, rheumatoid arthritis, sarcoidosis, systemic sclerosis, Sjogren’s syndrome, bullous pemphigoid, and immunoglobulin (IgG)4-related disease; medications, such as aspirin, nonsteroidal anti-inflammatory drugs, antimicrobials which can lead to drug rash with eosinophilia and systemic symptoms (DRESS) syndrome; malignancies, including solid tumors (lung, renal, colon) and Hodgkin and non-Hodgkin lymphoma; metabolic causes, such as adrenal insufficiency; immune deficiency, including hyper-IgE syndromes, Omenn syndrome, and Wiskott–Aldrich syndrome; and other more rare causes [3]. In order to investigate all aforementioned causes, the diagnostic algorithm includes travel history and parasite testing, including stool culture and antibodies for specific parasites (e.g., strongyloides), in the appropriate clinical context. Additional laboratory testing [e.g., IgE, troponin, autoimmune antibodies] and imaging tests (e.g., chest X-ray; electrocardiogram and cardiac echo; and chest, abdomen, and pelvis computed tomography) are guided by the patient’s history, symptomatology, and findings on clinical examination. For eosinophilic lung diseases, pulmonary function tests, bronchoscopy, and serology testing [e.g., aspergillus-specific IgE to evaluate for allergic bronchopulmonary aspergillosis (ABPA)] may be performed [10]. Evaluation of blood and/or bone marrow is characterized by meticulous investigation through serum B12 level, serum tryptase level, serum IgE level, FIP1 like-1 platelet-derived growth factor α-fusion gene (FIP1L1::PDGFRA) fusion by fluorescence in situ hybridization (FISH), or reverse transcription–polymerase chain reaction (RT-PCR) (either on blood sample or bone marrow biopsy), BCR::ABL1 fusion by FISH or RT-PCR (either on blood sample or bone marrow biopsy), T-cell receptor gene rearrangement by PCR (either on blood sample or bone marrow biopsy), immunophenotyping (either on blood sample or bone marrow biopsy), next-generation sequencing myeloid gene panel (either on blood sample or bone marrow biopsy), dysplasia and blast percentage (either on blood sample or bone marrow biopsy), bone marrow fibrosis, immunohistochemistry for CD117 and CD25 if serum tryptase is elevated (bone marrow biopsy), and standard karyotyping (bone marrow biopsy) [3,6,11,12,13,14]. The general diagnostic approach is depicted in a flow chart (Figure 1).

It is very challenging to decipher to what extent sustained elevated blood or tissue eosinophils will lead to organ failure. The existing data are insufficient to support a specific eosinophil count for the initiation of treatment in the absence of organ disease. Based on the literature, an absolute eosinophil count of 1.5 to 2 × 10^9^/L is the applied cut-off for initiating therapeutic modalities [3].

Traditional therapeutic options involve corticosteroids, characterized as first-line management unless FIP1L1/PDGFRA-positive; hydroxyurea, as a second-line therapeutic agent; vincristine, especially in children; other cytotoxic or immunomodulatory agents, such as rituximab; imatinib mesylate, as a first-line therapy for FIP1L1/PDGFRA-positive and myeloproliferative variants; and bone marrow transplants for FIP1L1/PDGFRA-positive and imatinib-resistant FIP1L1/PDGFRA-negative variants with disease progression despite standard of care therapies [15,16]. Over the past few years, extensive research efforts have dramatically improved our understanding of HES etiopathogenesis and led to actionable information for the diagnosis, prognosis, and therapeutic management of eosinophil disorders. In line with these, novel biologic agents, including mepolizumab, an anti-interleukin-(IL)5 monoclonal antibody, and benralizumab, a monoclonal antibody targeting IL-5 receptor alpha, have been shown to reduce disease flares and eosinophilic count and alleviate symptoms [5,17]. It is more than evident that the aforementioned beneficial role of IL-5-targeting monoclonal antibodies in the therapeutic management of HES stems from their mechanism of action. Mepolizumab directly binds to, and inactivates, circulating IL-5, while benralizumab, on the other hand, binds to the eosinophil cell membrane at the IL-5 receptor, preventing activation of the receptor, but simultaneously inducing apoptosis of the eosinophil by cross-binding with natural killer cells [18]. Mepolizumab was the first drug in over a decade to receive FDA approval for the treatment of HES without another identifiable non-blood-related cause [16,17,19]. More importantly, these targeted therapies not only present with noteworthy efficacy but also spare treatment-related toxicities. Nonetheless, real-life world evidence on both the safety and effectiveness of these agents is currently scarce and limited. To this end, we conducted a real-life study on biologic agents in patients with HES. The selection of specific biologic therapy was based on the physician’s decision, taking into account concomitant severe asthma and the patient’s preference.

## 2. Results

Eleven patients (*n* = 11) with a diagnosis of idiopathic HES were included in the study [M/F: 6/5, median age: 54 (95% CI: 38.2 to 68.5), smokers/never smokers: 5/6]. Asthma was present in the majority of them (*n* = 8, 72.7%); four patients (*n* = 4, 36.4%) presented with eosinophilic pleural effusions, two patients (*n* = 2, 18.2%) with cardiac arrhythmias, and one with bilateral eyelid angioedema. Eight patients (72.7%) were treated with mepolizumab (300 mg/month) and three (27.3%) with benralizumab (30 mg/4 weeks). The median values of eosinophils at baseline and 12 months after the initiation of biologic treatment were 3000 (95% CI: 2172 to 11,365) K/μL and 50 (95% CI: 3 to 190) K/μL, respectively, *p* = 0.0002. All patients with concomitant asthma (*n* = 8)—five of them were treated with mepolizumab (300 mg/month) and three of them were treated with benralizumab (30 mg/4 weeks)—experienced elimination of asthma flares, asthma control [Asthma Control Questionnaire (ACQ) < 0.75], and functional improvement (mean ΔFEV1: 857 ± 594 mL). No significant difference was observed between the two groups of applied biologic agents—mepolizumab and benralizumab—with regard to outcomes. HES flares had been defined as HES-attributable worsening of clinical symptomatology or blood eosinophil counts requiring an escalation in therapy. With regard to the longitudinal use of oral corticosteroids, we noticed that 82% of the patients achieved elimination (*p* = 0.0001), while the two patients that remained on maintenance oral corticosteroids achieved a reduction in dosing (Table 1, Figure 2). It is also remarkable that all patients improved in terms of forced expiratory volume in 1 s (FEV_1_) and forced vital capacity (FVC) on the treatment, despite only one subset of them having been diagnosed with asthma. A diagnosis of asthma was not established in the other subgroup despite the functional improvement in spirometry. Importantly, none of our patients experienced any permanent organ damage and/or dysfunction attributable to tissue eosinophilia. No treatment-related adverse events were noticed. All patients were assessed through clinical examination, laboratory tests, vital signs, or electrocardiographic results for excluding major safety concerns such as hypereosinophilic flares, lymphocytopenia, renal failure, hepatitis, pneumonia, polyneuropathy, or cardiac arrhythmia, or adverse events such as headache, fatigue, peripheral edema, rash, or arthralgia. In addition, none of our patients experienced adverse events associated with the longitudinal use of oral corticosteroids.

## 3. Discussion

To our knowledge, this was one of the largest real-world studies including a considerable number of highly characterized patients with idiopathic HES treated with anti-eosinophilic biologic agents so far, considering disease rarity [20]. Based on a diagnostic algorithm of eosinophilic disorders, all patients enrolled in our cohort underwent mandatory first-line investigations, as well as further investigations according to patient history and clinical manifestations [6]. Infectious diseases, as well as drug-related hypereosinophilia, were ruled out. Subsequently, real-time quantitative PCR, or FISH and flow cytometry lymphocyte phenotyping, were performed in assessing FIP1L1-PDGFRA gene fusion, rare mutations and rearrangements, and clonal or aberrant T-cell populations in order to exclude clonal bone marrow disease [21]. Based on the fact that HES is a systemic disease and fully dependent on levels of eosinophils, including concomitant coagulation impairment, inflammation. and tissue damage, organ involvement was specifically addressed. The respiratory system, skin, gastrointestinal system, heart vessels, and nervous system were thoroughly assessed.

With regard to therapeutic management, we demonstrated that biologic agents in patients with idiopathic HES are both safe and effective, sparing the toxicity of immunosuppressive agents, including myelosuppression, opportunistic infections, malignancies, or further side effects of long-term corticosteroid use such as hypertension, diabetes mellitus, osteoporosis, myopathy, gastritis, and glaucoma. Our findings are in line with large-scale randomized controlled trials showing the superiority of mepolizumab compared to placebo in reducing disease flares and use of oral corticosteroids. We are also in accordance with a smaller phase 2 trial showing the efficacy of benralizumab in decreasing blood and tissue eosinophilia with no serious toxic effects in patients with severe, treatment-refractory hypereosinophilic syndrome [17,19]. It is important to know that in our country, Greece, mepolizumab is indicated for patients with HES other than asthma and benralizumab is administrated in patients with HES and concomitant severe asthma. Both of them are fully reimbursed following approval by national health committees.

Despite the exponential increase in our understanding of the therapeutic management of eosinophil disorders, the duration of administration of biologic agents targeting eosinophils is still puzzling, taking into account their high cost reimbursed by health systems. Data derived from the literature show that most patients are prone to a baseline level of blood eosinophils and increased risk of exacerbations following the cessation of biologic agents in the context of severe asthma treatment [22,23,24]. As a consequence, the sudden cessation of biologic treatment in patients with HES could lead to a loss of disease control. Recently, Soendergaard et al., in the OPTIMAL study, suggested that gradual down-titration of anti-IL-5 biologics in severe asthma could be a more appropriate approach to evaluate whether a reduction in or cessation of treatment could be possible, raising concerns for personalized dosing intervals in the future [25]. These data need further validation on a longitudinal basis. In addition, further caution should be applied for potential treatment cessation in disorders lying within the spectrum of hypereosinophilia, as excess eosinophilia requires continuous and more aggressive therapeutic (anti-eosinophilic and/or cytotoxic) approaches.

The number of patients included in our study could be a limitation, but idiopathic HES is a rare entity (10/1,000,000) with scarce real-life data in the literature, so our study could really contribute to this field. In particular, our data represent the largest real-life data on the management of highly characterized HES patients in Greece, and one of the largest published so far in Europe. Thus, we believe that our data add major and valuable knowledge on the effectiveness and safety of biologics in hypereosinophilic disorders in real-life clinical settings.

## 4. Materials and Methods

We recorded consecutive patients between 10 January 2019 and 1 January 2023 who presented in our Respiratory Department of the University Hospital of Patras, Greece, received a multidisciplinary diagnosis of idiopathic hypereosinophilic syndrome, and were treated with anti-eosinophilic biologic agents from diagnosis until the present. Diagnosis was based on clinical, laboratory, radiological, and histological examination. Secondary causes of eosinophilia, such as infections, particularly tissue-invasive parasites; allergy or atopy and hypersensitivity conditions; drug reactions; collagen–vascular disease (e.g., eosinophilic granulomatosis with polyangiitis, granulomatosis with polyangiitis, systemic lupus erythematosus, IgG4-related disease) pulmonary eosinophilic diseases (e.g., idiopathic acute or chronic eosinophilic pneumonia, allergic bronchopulmonary aspergillosis); allergic gastroenteritis (with associated peripheral eosinophilia); immune deficiency (e.g., hyper-IgE syndromes); and metabolic conditions such as adrenal insufficiency, were excluded, and an evaluation of primary bone marrow disorders was conducted. All patients had a negative immunological profile including *p*, c-ANCA, cyclic citrullinated peptides (CCPs), ANA and extractable nuclear antigen (ENA) panel, normal aspergillus-specific IgE, negative stool culture, normal examination of blood smear, normal serum B12, and tryptase, as well as negative FIP1L1-PDGFRA. A full-body computed tomography scan, pulmonary function testing, bronchoscopy, and hematological consult were included in the work-up of all patients. The diagnostic criteria used for idiopathic HES stem from the definition of the disorder and include hypereosinophilia (>1500 K/μL), related organ dysfunction, and exclusion of secondary causes of hypereosinophilia. Data collection and analysis was approved by the Institutional Review Board and the Local Ethics Committee (protocol number: 19281/05-JUL-2023). Data collection included demographics and history for previous administration of corticosteroids, exacerbations, and comorbidities. Informed consent was obtained from all individual participants included in the study.

## 5. Conclusions

In conclusion, following the exclusion of eosinophilia due to secondary causes, the diagnostic approach to primary eosinophilia necessitates a combination of a plethora of assessments. Our real-life clinical data support the effectiveness and safety of biologic agents in highly characterized idiopathic HES through the elimination of flares, a reduction in blood eosinophilia, and their steroid-sparing effects. Future real-world evidence from larger registries aiming to identify the safety and efficacy profiles of anti-eosinophilic biologic agents in HES is greatly anticipated.

## Figures and Tables

**Figure 1 pharmaceuticals-18-00543-f001:**
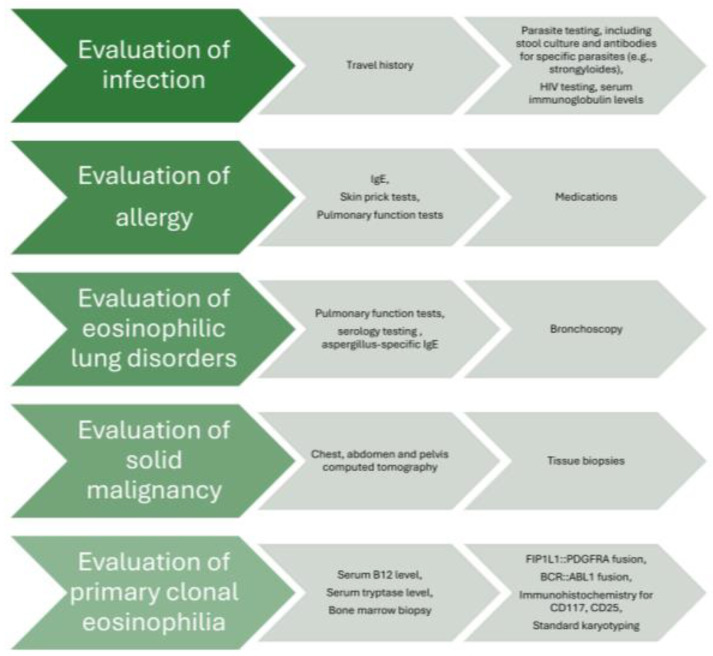
Flow chart of diagnostic approach in idiopathic HES.

**Figure 2 pharmaceuticals-18-00543-f002:**
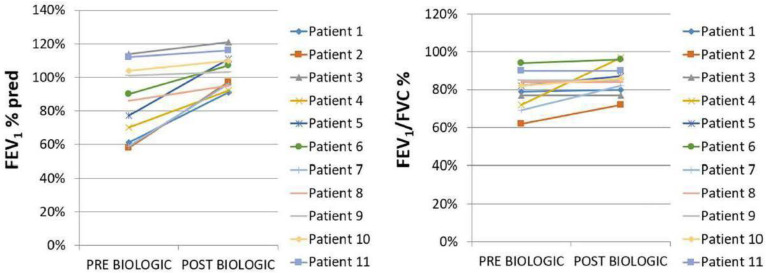
FEV1% pred and FEV1/FVC% tracings pre (baseline) vs. post (12 months after initiation of biologic agent) biologic treatment, *p* = 0.04 and *p* = 0.14, respectively.

**Table 1 pharmaceuticals-18-00543-t001:** Characteristics of enrolled patients pre- and 12 months after initiation of biologic treatment (BT).

Characteristics	Pre-BT	Post BT
Oral corticosteroids: prednisone (longitudinal use)	11 (100%)–20 mg/d	2 (18%)–7.5 mg/d
Median value of eosinophils (K/μL) (95%CI)	3000 (2172 to 11,365)	50 (3 to 190)
HES flares	11 (100%)	0 (0%)
Asthma control (ACQ < 0.75)	0 * (0%)	8 * (100%)
ΔFEV1 ± SD	NA	857 ± 594 mL

* Asthma was present in 8 patients with HES.

## Data Availability

Data are available on request. O.P. and A.T. have full access to the data and are the guarantors for these data.

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
