# Peer review of "Biologic Agents in Idiopathic Hypereosinophilic Syndrome†"

_pharmaceuticals, 2025, doi:10.3390/ph18040543_

Round 1

Reviewer 1 Report

Comments and Suggestions for Authors

Papaioannou et al. have presented a real-world analysis of biologic therapy in idiopathic hypereosinophilic syndrome (HES). Although the study is not highly novel, real-world data on the effectiveness of targeted therapies in rare diseases like HES are valuable. These findings contribute to the growing body of evidence supporting IL-5/IL-5R blockade as a safe and effective option, reinforcing its role in disease management.

I have a few suggestions that could further strengthen the manuscript:

  • It would be helpful to discuss what happens when IL-5 antibodies are withdrawn. For how long are HES patients expected to continue IL-5-blocking therapy? Is there a risk of symptom recurrence or eosinophilia rebound after stopping treatment? Are there any studies looking at this in long-term?
  • A brief explanation of the mechanism of action of IL-5-targeting monoclonal antibodies would provide clarity. How targeting IL-5 or its receptor can improve HES.
  • The manuscript states that IL-5/IL-5R mAbs reduce immunosuppressive-driven toxicities, but it would be useful to specify what these toxicities are.
  • Authors report that no treatment-related adverse events were observed in patients receiving the biologics. However, it would strengthen the manuscript if you specify what types of adverse events were assessed.
  • It would be beneficial to acknowledge the potential drawbacks of these targeted therapies, such as their high cost, etc., to provide a more balanced discussion.

Author Response

We would like to thank the reviewers for their valuable comments and the fast peer-review process. Comments obviously stem from experts in the field and therefore we are grateful, as they really helped us to improve the quality of our manuscript. 

Our point-by-point reply to reviewers follows:

Reviewer 1:

Papaioannou et al. have presented a real-world analysis of biologic therapy in idiopathic hypereosinophilic syndrome (HES). Although the study is not highly novel, real-world data on the effectiveness of targeted therapies in rare diseases like HES are valuable. These findings contribute to the growing body of evidence supporting IL-5/IL-5R blockade as a safe and effective option, reinforcing its role in disease management.

I have a few suggestions that could further strengthen the manuscript:

  • It would be helpful to discuss what happens when IL-5 antibodies are withdrawn. For how long are HES patients expected to continue IL-5-blocking therapy? Is there a risk of symptom recurrence or eosinophilia rebound after stopping treatment? Are there any studies looking at this in long-term?

AU: Thank you for your really interesting comment. We added the following comment in discussion section:

“Data derived from the literature showed that most patients are prone to the baseline level of blood eosinophils and increased risk of exacerbations following cessation of biologic agent in the context of severe asthma treatment (21-23). As a consequence, sudden cessation of biologic treatment in patients with HES could lead to loss of disease control. Recently, Soendergaard et al, in OPTIMAL study, suggested that gradual down-titration of anti-IL-5 biologics in severe asthma could be a more appropriate approach to evaluate whether reduction or cessation of treatment could be possible raising concerns for personalized dosing intervals in the future (24). These data need further validation on a longitudinal basis. In addition, further caution should be applied for potential treatment cessation in disorders lying within the spectrum of hypereosinophilia, as excess eosinophilia require continuous and more aggressive therapeutic (anti-eosinophilic and/or cytotoxic) approaches.”

  • A brief explanation of the mechanism of action of IL-5-targeting monoclonal antibodies would provide clarity. How targeting IL-5 or its receptor can improve HES.

AU: Thank you for your comment. We added the following in introduction section:

It is more than evident that the aforementioned beneficial role of IL-5-targeting monoclonal antibodies in therapeutic management of HES stems from their mechanism of action. Mepolizumab directly binds to, and inactivates, circulating IL-5, while benralizumab, on the other hand, binds to the eosinophil cell membrane at the IL-5 receptor, preventing activation of the receptor, but simultaneously inducing apoptosis of the eosinophil by cross-binding with natural killer cells (18).”

  • The manuscript states that IL-5/IL-5R mAbs reduce immunosuppressive-driven toxicities, but it would be useful to specify what these toxicities are.

AU: Thank you for your comment. We apologize for not reporting this. We added the following in discussion section:

“…sparing toxicity of immunosuppressive agents, including myelosuppression, opportunistic infections, malignancies or further side effects of long-term corticosteroid use such as hypertension, diabetes mellitus, osteoporosis, myopathy, gastritis and glaucoma.”

  • Authors report that no treatment-related adverse events were observed in patients receiving the biologics. However, it would strengthen the manuscript if you specify what types of adverse events were assessed.

AU: Thank you for your comment. We apologize for not reporting this. We added the following in results section:

“All patients were assessed through clinical examination, laboratory tests, vital signs, or electrocardiographic results for excluding major safety concerns such as hypereosinophilic flare, lymphocytopenia, renal failure, hepatitis, pneumonia, polyneuropathy and cardiac arrhythmia or adverse events such as headache, fatigue, peripheral edema, rash or arthralgia”

  • It would be beneficial to acknowledge the potential drawbacks of these targeted therapies, such as their high cost, etc., to provide a more balanced discussion.

AU: Thank you for your comment. We added the following in discussion section:

“Despite the exponential increase in our understanding of therapeutic management of eosinophil disorders, duration of administration of biologic agents targeting eosinophils is still puzzling taking into account their high cost reimbursed by health systems”

Reviewer 2 Report

Comments and Suggestions for Authors

Authors have conducted a real-life study for biologic agents in patients with HES. Although the study appears interesting, the following points may be noted: 

  1. The rationale of inclusion of different biologics for the study needs to be mentioned.
  2. The difference between the effect of the two biologics would be interesting and needs to be commented upon.
  3. It would be useful if a flow chart can be included to list out the various investigations performed and their rationale while HES as a disease of exlusion has been established.

Author Response

We would like to thank the reviewers for their valuable comments and the fast peer-review process. Comments obviously stem from experts in the field and therefore we are grateful, as they really helped us to improve the quality of our manuscript. 

Our point-by-point reply to reviewers follows:

Reviewer 2:

Authors have conducted a real-life study for biologic agents in patients with HES. Although the study appears interesting, the following points may be noted: 

  1. The rationale of inclusion of different biologics for the study needs to be mentioned.

AU: Thank you for your comment. As mentioned in the discussion, in our country, Greece, mepolizumab is indicated for patients with HES other than asthma and benralizumab is administrated in patients with HES and concomitant severe asthma. Both of them are fully reimbursed following approval by national health committees.

In order to be more concise with regards to rationale of inclusion of different biologics we added the following in introduction section:

“The selection of specific biologic therapy was based on the physician’s decision, taking into account concomitant severe asthma and patient’s preferences.

  1. The difference between the effect of the two biologics would be interesting and needs to be commented upon.

AU: Thank you for your comment. We added the following in introduction section:

“It is more than evident that the aforementioned beneficial role of IL-5-targeting monoclonal antibodies in therapeutic management of HES stems from their mechanism of action. Mepolizumab directly binds to, and inactivates, circulating IL-5, while benralizumab, on the other hand, binds to the eosinophil cell membrane at the IL-5 receptor, preventing activation of the receptor, but simultaneously inducing apoptosis of the eosinophil by cross-binding with natural killer cells (18).”

  1. It would be useful if a flow chart can be included to list out the various investigations performed and their rationale while HES as a disease of exlusion has been established.

AU: Thank you for your suggestion. We added the flow chart as Figure 1.

Reviewer 3 Report

Comments and Suggestions for Authors

This is a very small clinical report that includes 11 patients only.  Although HES is a rare disease, more n numbers are needed.  In addition, the result of 2 biological interventions are presented, descriptively, stated as being beneficial, statistical analysis is necessary to subjectify the result. 

Author Response

We would like to thank the reviewers for their valuable comments and the fast peer-review process. Comments obviously stem from experts in the field and therefore we are grateful, as they really helped us to improve the quality of our manuscript. 

Our point-by-point reply to reviewers follows:

Reviewer 3:

This is a very small clinical report that includes 11 patients only.  Although HES is a rare disease, more n numbers are needed.  In addition, the result of 2 biological interventions are presented, descriptively, stated as being beneficial, statistical analysis is necessary to subjectify the result.

AU: Thank you for your comments. We understand your concerns about the number of patients included, but idiopathic HES is a rare entity (10/1,000,000) with scarce real-life data in literature, so our study could really contribute to this field. In particular, our data represents the largest real-life data on the management of highly-characterized HES patients in Greece, and one of the largest published, so far, in Europe. Thus, we believe that our data add major and valuable knowledge on the effectiveness and safety of biologics in hypereosinophilic disorders in the real-life clinical setting.

With regards to statistical analysis, we provided p-values for:

Median values of eosinophils at baseline and 12 months after initiation of biologic treatment were 3000 (95% CI: 2172 to 11365) K/μL and 50 (95% CI: 3 to 190) K/μL, respectively, p=0.0002.

With regards to longitudinal use of oral corticosteroids, we noticed that 82% of the patients achieved elimination (p=0.0001).

Figure 2. FEV1% pred and FEV1/FVC% Tracings pre (baseline) vs post (12 months after initiation of biologic agent) biologic treatment, p=0.04 and p=0.14, respectively.

Sincerely,

Argyrios Tzouvelekis MD, MSc, PhD

Professor of Respiratory Medicine

Head Department of Respiratory Medicine

University of Patras, Greece

Associate Professor Adjunct, PCCSM, Yale School of Medicine, USA

atzouvelekis@upatras.gr, argyris.tzouvelekis@gmail.com

Round 2

Reviewer 2 Report

Comments and Suggestions for Authors

Thank you for the efforts put into addressing the comments. However, some comments have been misinterpreted. 

  1. The difference between the effect of the two biologics pertaining the result of the study would be interesting. Is there any difference in results between the two biologics?
  2.  Flow chart should list out the various individual investigations ( tests) performed for ruling out each possible etiology. 

Author Response

We would like to thank the reviewers for their valuable comments and the fast peer-review process. They really helped us to improve the quality of our manuscript. 

Our point-by-point reply to reviewers follows:

Reviewer 2:

Thank you for the efforts put into addressing the comments. However, some comments have been misinterpreted.

The difference between the effect of the two biologics pertaining the result of the study would be interesting. Is there any difference in results between the two biologics?

Flow chart should list out the various individual investigations (tests) performed for ruling out each possible etiology.

AU: Thank you for your clarifications. We added the following in results section:

“No significant difference was observed between the two groups of applied biologic agents -mepolizumab and benralizumab- with regards to outcomes.”

Flow chart was modified, as requested.

Reviewer 3 Report

Comments and Suggestions for Authors

The authors added the statistical analysis and revised the manuscript accordingly.  The small n number problem is a whole different issue that should be addressed once the cases are accumulated in the continued clinical followup.

Author Response

We would like to thank the reviewers for their valuable comments and the fast peer-review process. They really helped us to improve the quality of our manuscript. 

Our point-by-point reply to reviewers follows:

Reviewer 3:

The authors added the statistical analysis and revised the manuscript accordingly.  The small n number problem is a whole different issue that should be addressed once the cases are accumulated in the continued clinical follow-up.

AU: Thank you for your comment.

We believe that the number of patients included in our study is clearly stated as limitation, but we explained that idiopathic HES is a rare entity with scarce real-life data in literature, so our study could really contribute to this field. In particular, our data represents the largest real-life data on the management of highly-characterized HES patients in Greece, and one of the largest published, so far, in Europe. It is also mentioned in conclusion that future real-world evidence from larger registries aiming to identify the safety and efficacy profiles of anti-eosinophilic biologic agents in HES are greatly anticipated.
